# Elevated growth and biomass along temperate forest edges

Luca L. Morreale [1,2 ✉], Jonathan R. Thompson[2], Xiaojing Tang [1], Andrew B. Reinmann[3,4,5] & Lucy R. Hutyra[1]

Fragmentation transforms the environment along forest edges. The prevailing narrative, driven by research in tropical systems, suggests that edge environments increase tree mortality and structural degradation resulting in net decreases in ecosystem productivity. We show that, in contrast to tropical systems, temperate forest edges exhibit increased forest growth and biomass with no change in total mortality relative to the forest interior. We analyze >48,000 forest inventory plots across the north-eastern US using a quasi-experimental matching design. At forest edges adjacent to anthropogenic land covers, we report increases of 36.3% and 24.1% in forest growth and biomass, respectively. Inclusion of edge impacts increases estimates of forest productivity by up to 23% in agriculture-dominated areas, 15% in the metropolitan coast, and +2% in the least-fragmented regions. We also quantify forest fragmentation globally, at 30-m resolution, showing that temperate forests contain 52% more edge forest area than tropical forests. Our analyses upend the conventional wisdom of forest edges as less productive than intact forest and call for a reassessment of the conservation value of forest fragments.

[1] Department of Earth & Environment, Boston University, Boston, MA, USA. [2] Harvard Forest, Harvard University, Petersham, MA, USA. [3] Environmental Science Initiative, CUNY Advanced Science Research Center, New York, NY, USA. [4] Graduate Program in Earth and Environmental Sciences and Biology, CUNY Graduate Center, New York, NY, USA. [5] Department of Geography and Environmental Sciences, Hunter College, New York, NY, USA. ✉email: lmorreal@bu.edu

Deforestation is a pervasive consequence of land-use change[1] and is impactful not just due to what is lost, but also due to its effects on the forest fragments that remain. Forest fragmentation is globally ubiquitous, with over 70% of forests located less than 1 km from a non-forest edge[2]. Fundamental constraints on forest growth[3,4] and carbon cycling are altered near edges relative to interior forests[5,6], with increases in light availability, temperature, wind, and reactive nitrogen deposition, as well as altered water availability[7,8]. While fragmentation occurs across biomes, reported effects of these perturbations on higher-order dynamics in fragmented forests (i.e., structure, composition, function, and mortality) have largely focused on tropical ecosystems, where sharp increases in mortality and long-term forest degradation are reported at the forest edge[9–13]. Expanded analyses suggest significant reductions in tropical ecosystem net carbon sequestration and, more broadly, the terrestrial carbon sink[10,11,14]. However, environmental controls on temperate forests differ from the tropics, and temperate forest fragmentation studies are both fewer and more limited in scale, c.f. [15,16]. Temperate forest edges have similar microclimatic differences, but contrasting biomass and productivity responses, emphasizing a need for a better understanding of edge ecosystems in non-tropical biomes[6,15,17,18].

Here, we offer a large-scale estimation of fragmentation impacts on temperate forest growth and structure along forest edges, with broader implications for global evaluation of fragmented forests. Hereafter, we use the term edge to refer to forest area bounded, in part, by a non-forest land cover and, conversely, interior as a designation of forest area bounded fully by forest. We report differences in tree basal area (BA; a metric of forest structure, strongly correlated with biomass), BA increment (BAI; a measure of forest growth), tree mortality, and average stem density and diameter, between the forest edge (edge plots; <15 m from a non-forest land cover) and forest interior (interior plots; nonadjacent to non-forest land cover). We show that the temperate forest edges within our study area exhibit dramatically increased growth, tree stem density, and total BA, with negligible changes in mortality. We then scale these results to estimate regional increases in forest growth attributable to the distinct forest edge environment. Finally, we place our results in context of global patterns of forest fragmentation.

## Results and discussion

**Distinct characteristics of forest edges.** To examine forest edges in the northeastern US, we used inventory data from the US Department of Agriculture Forest Inventory and Analysis (FIA) program. The FIA program has established permanent, fixed-area (675 m²), forest plots in a hexagonal grid across the United States[19]. This national forest inventory includes measurements of tree size, growth, and land use; re-measuring every 5–7 years in our study area. Using >48,000 FIA plots distributed throughout 20 northeastern US states (Supplementary Fig. 1), we compared structural and growth dynamics along temperate forest edges to those of interior forests. Individual tree measurements are collected within four fixed-radius subplots (168.7 m² area) with a fixed orientation; subplot characteristics are recorded even if the subplot contains partially forested or non-forest area. We leverage partially forested subplots to identify forest edges within the FIA database.

Using a quasi-experimental statistical matching framework followed by a generalized linear model (GLM) regression analysis, we compared BA, BAI, and tree mortality on FIA subplots that are adjacent to a non-forest land cover, to matched subplots within the forest interior. Matching approximates an experimental design where control plots (interior) were selected based

on similarity to the treatment plots (edge) in relation to confounding predictors (light, water, temperature, nitrogen deposition, and forest type; Supplementary Fig. 5)[20]. We report the results from GLM regression models as percent differences with significance derived from Wald tests on regression coefficients and we include Nagelkerke Pseudo-$R^2$ from the most parsimonious models as a goodness-of-fit metric[21]. Detailed descriptions of plot filtering, statistical matching, GLM selection and analysis are provided in the Methods section.

Edges come in many forms. Natural edges exist as both transitions in growing conditions (e.g., forest–grassland ecotones, and wetlands) and sharp boundaries (e.g., lakes, rivers, and geologic features) with variable effects on forest growth. In contrast, anthropogenic edges often exist as abrupt transitions in areas that were once fully forested (e.g., agricultural fields, roads, and developments). Average BAI along anthropogenically formed edges is 36.3% greater ($p < 0.001$; $R^2 = 0.149$) than interior forest, while BAI along all edges (encompassing anthropogenic, natural, and unspecified edges) is 24.1% greater ($p < 0.001$; $R^2 = 0.153$) than interior forest (Fig. 1). BA exhibits smaller differences, but the same trend: anthropogenic edges have 21.0% greater ($p < 0.001$; $R^2 = 0.059$) BA and along all edges BA is 13.9% greater ($p < 0.001$; $R^2 = 0.069$) than the forest interior. Notably, our analyses exclude trees smaller than 12.7 cm in diameter. Given that densities of small diameter woody vegetation are typically higher along forest edges[6], it follows that the differences in BA and BAI between edge and interior forests observed here represent a conservative estimate.

There are just three pathways to increased BA in edge forests: more trees, larger trees, or some combination thereof. We find no significant difference in the average tree diameter between the forest edge and interior, even when comparing with only anthropogenic edges. In contrast, by averaging individual tree measurements within each subplot, we find a mean increase of 58 trees per hectare ($p < 0.001$) across all edges as compared with the forest interior (Fig. 2). Along anthropogenic edges, the difference increases to 82.6 additional trees per hectare ($p < 0.001$), which is consistent with the observed patterns of BA in all versus anthropogenic edges.

Along tropical edges, the primary driver of decreased productivity is heightened tree mortality, frequently attributed to increased impacts of wind, lianas, and more frequent droughts[22]. In contrast, we find no significant differences in biogenic mortality between edge and interior forests (Supplementary Fig. 3b). Within our study area, the largest cause of mortality in forests is anthropogenic removals[23]. While we do find a statistically significant ($p < 0.001$) increase in anthropogenic removals in both edge groups compared to the interior (Supplementary Fig. 3c), there is no difference in overall total mortality (Supplementary Fig. 3a). Given the prevalence of forest management in this region, we performed a robustness test of our main result to quantify any potential impacts of harvesting. We withheld all plots that had a record of tree removal ($n = 3642$) within the FIA inventory and found no changes in the overall pattern between edge and interior in either BA or BAI.

Tree species composition mediates forest response to anthropogenic environmental perturbations[24]. Individual species responses to altered energy and biogeochemical inputs at the edge can vary due to climatic tolerance and successional characteristics[25]. Therefore, we quantified differences in structure and growth responses to edges by species composition groups[23] (Fig. 1). In most compositional groups, BAI increases significantly at all forest edges, but with varying magnitudes: Northern Pines—Hemlock forests exhibit the smallest increase in BAI, 16.9% ($p < 0.001$); Oak—Pine forests have the largest, 32.5% ($p < 0.001$). The effect size increases across almost all compositional groups

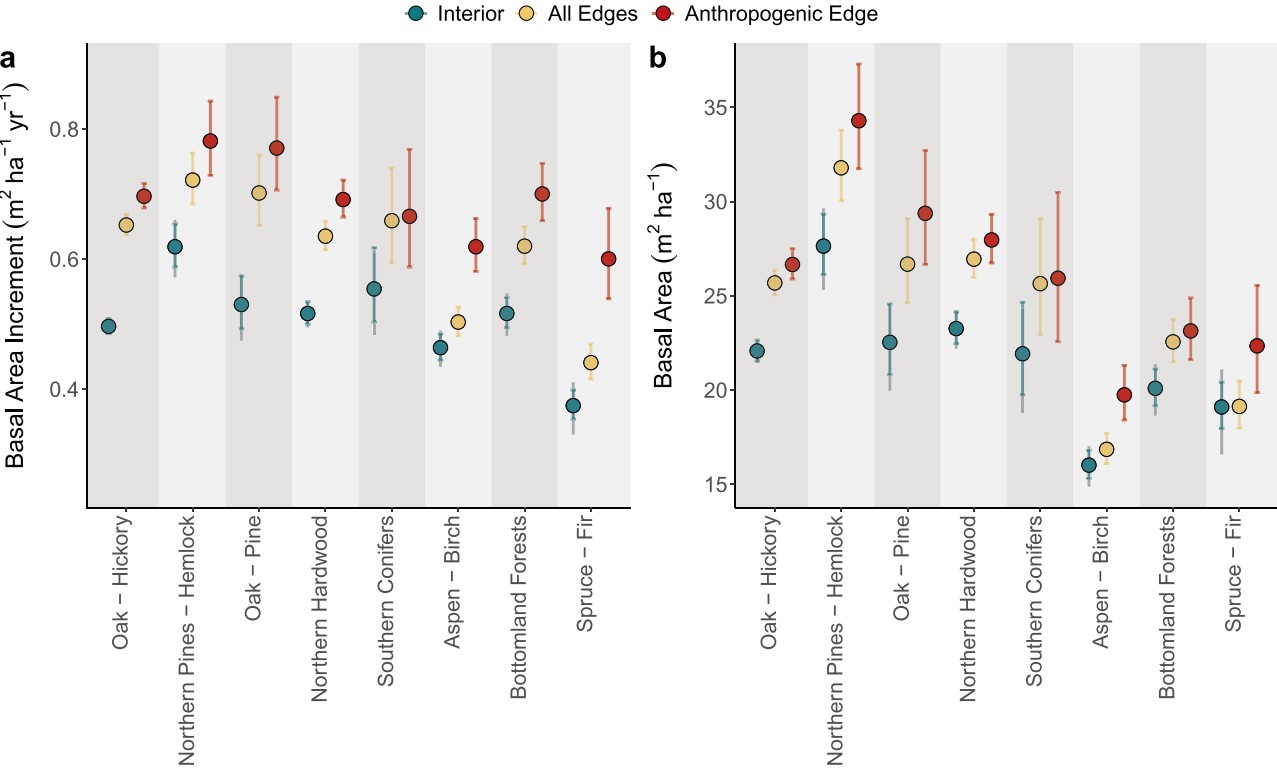

**Fig. 1 Forest edges have elevated growth and basal area.** BAI (**a**) and BA (**b**) show the average marginal effects of edge-class and forest-type from GLM outputs. Results are presented in Interior, All edges, and Anthropogenic edge groups and ordered by forest type abundance (Supplementary Fig. 5). Interior and All Edge groups have $n = 6607$ independent subplots, anthropogenic edges have $n = 4327$ independent subplots. Data are presented as the mean marginal effects with inner error bars showing 95% confidence intervals on the marginal effects; outer error bars on interior group are for comparison with anthropogenic edges.

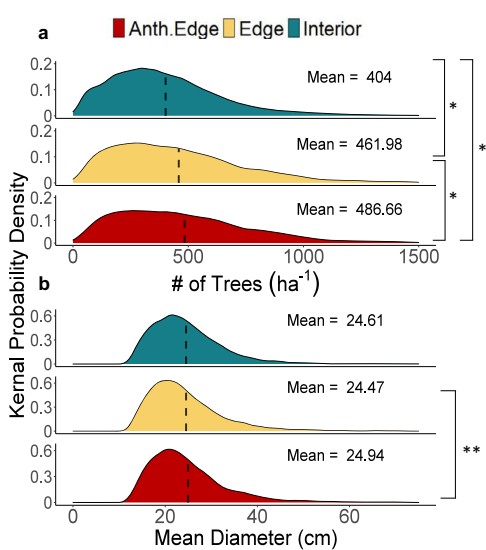

**Fig. 2 Temperate forest edges have higher mean stem density than the forest interior but exhibit no difference in mean tree diameter.**
**a** Distributions of mean subplot stem density (# of trees per hectare).
**b** Distributions of mean subplot tree diameter (diameter in centimeters). Dashed lines show mean values of all subplots within each edge class. Asterisks denote significance (*$p < .00001$; **$p = 0.0078$) as calculated with two-sided pairwise *t* tests using a Bonferroni adjustment. Interior and all edge groups have $n = 6607$ independent subplots, anthropogenic edges have $n = 4327$ independent subplots.

when comparing BAI specifically along anthropogenic edges with forest interiors. Of the eight forest type groups, only the Southern Conifers group has no statistically significant difference in BAI. The increase in BAI ranges from 25.5% ($p < 0.001$) in Northern Pines—Hemlock, to 67.7% in Spruce—Fir. The Oak—Hickory group exhibits 41.1% ($p < 0.001$) higher tree growth at anthropogenic edges than the forest interior, an effect >28% larger than when all edges are pooled. Interior-to-edge enhancements of BA are smaller than BAI, but five compositional groups have significantly greater BA along edges: Oak—Hickory (16.5%; $p < 0.001$), Northern Hardwood (16.1%; $p < 0.001$), Northern Pines—Hemlock (15.1%; $p < 0.001$), Oak—Pine (18.5%; $p < 0.001$), and Bottomland Forests (12.5%; $p < 0.001$). When comparing anthropogenic edges with the interior, the effect is again stronger, and five compositional groups exhibit significant increases in edge BA. Of these groups, Aspen—Birch have the largest increase in BA (31.7%; $p < 0.001$); Northern Hardwoods have the smallest (19.5%; $p < 0.001$).

**Estimating the regional impact of enhanced growth.** To scale the edge impacts on growth across our study area, we coupled the results from the GLM regression analysis with a land-cover map[26] and a forest-type map[27]. We aggregate our results to ecoregions, geographic areas that are ecologically and climatically similar, to account for mismatches in spatial resolution between our gridded inputs[28,29]. For these analyses, we focused on the effects of anthropogenic edges. The increases in growth and biomass we observe at temperate forest edges are greatest adjacent to anthropogenic edges and are evidence of a largely unrecognized impact of the ongoing process of forest fragmentation. Large variability was observed in fragmentation patterns across our

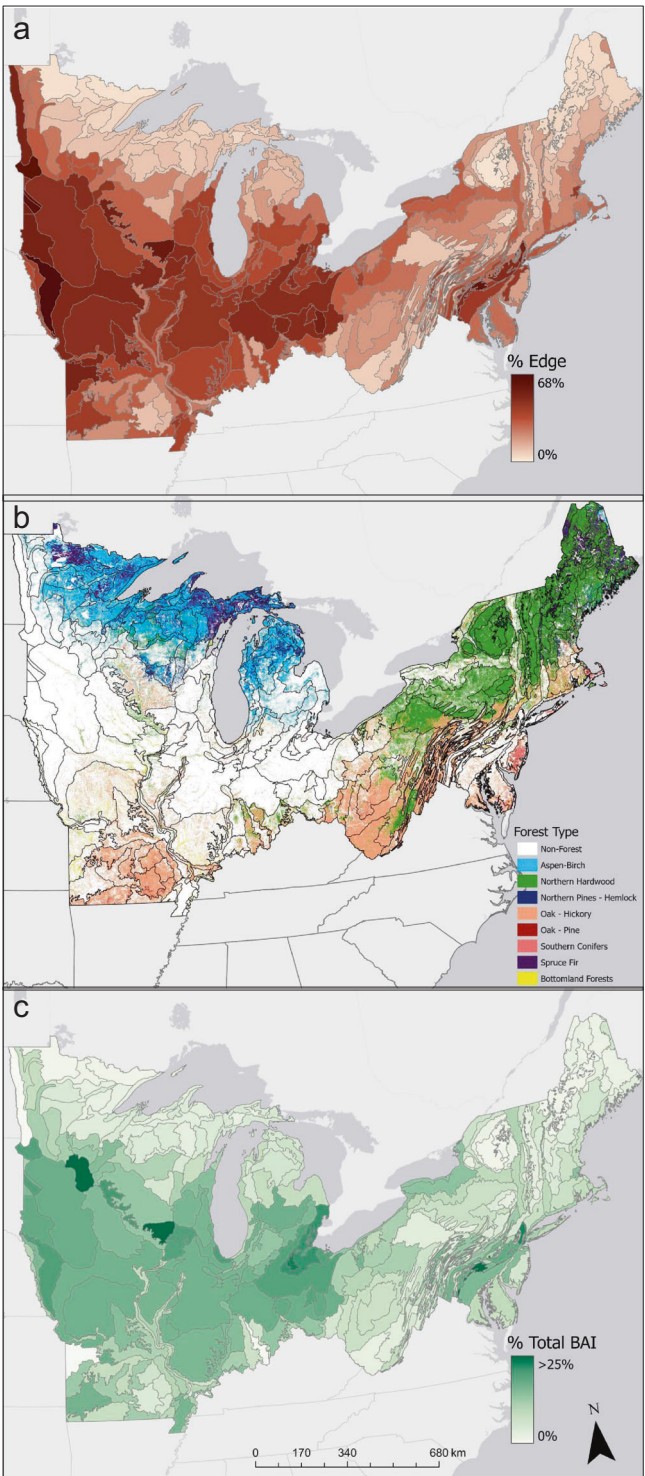

**Fig. 3 Edges increase productivity in temperate forests. a** The percent of forest area within 30 m of an anthropogenic edge within each ecoregion. **b** Spatial distribution of aggregated forest types used in study. **c** The percent increase in ecoregion total BAI attributable to elevated growth at anthropogenic edges.

edge with ecoregion BAI differences to quantify the effect of edges on overall forest productivity. We estimated the total increase in annual BAI within each ecoregion associated with increased growth at anthropogenic forest edges (Supplementary Fig. 5). Estimates determined that elevated BAI found at anthropogenic forest edges represents a >6% increase in total forest growth across the entire region (Fig. 3c). The BAI response varied across our study domain; increases in forest growth range from 23% increase in agricultural-dominated areas (region shown in Supplementary Fig. 6b), a 2% increase in the least-fragmented northern regions (region shown in Supplementary Fig. 6c), and a 15% increase within the metropolitan east coast (region shown in Supplementary Fig. 6d).

Our findings contrast with the conventional narrative based on tropical forest studies, that forest edges decrease net forest productivity and, consequently, lower forest aboveground carbon storage. Temperate and tropical forests have distinct ecologies and climate; it follows that similar perturbations can have markedly different effects. The absence of any increase in tree mortality, as repeatedly observed in tropical forest edges, suggests that temperate forest edges are less wind-threatened and less sensitive to the elevated temperatures and water stress that occur along all forest edges. Rather, increases in radiation may release the most-limiting biogeochemical constraints on temperate forests (temperature and light)[3,6,18]. The growth response is almost certainly related to greater light availability, which affects tree canopy architecture and can increase forest leaf area index and, in turn, stimulate productivity[18,30].

**The global extent of forest fragmentation.** Comparison of our results and those of previous tropical studies is complicated by differences in land-use history, specifically the time since edge creation. Forests in our study region and, more broadly, the temperate forest biome have undergone centuries of deforestation, forest transitions, and fragmentation. Some forest edges included in our study have existed for decades. However, research on newly created edges in this region has shown large growth increases in remaining trees, without associated increases in mortality, immediately following edge creation[31]. Given that abrupt formation of edges can expose the previously intact forest to secondary disturbances, individual tree characteristics, including height, drought tolerance, and rooting depth, may determine whether the cascading perturbations induce mortality. Shorter, more wind-firm trees, prevalent in temperate forests, may not experience altered biogeochemical conditions only as negative perturbations and, instead, are more likely to be advantaged by increased resource availability. In contrast, the taller trees found in temperate forests of the Pacific Northwestern US, in which fragmentation patterns are characterized by deforestation and clear-cut timber harvests, might exhibit a similar initial mortality response to tropical forests[29]. However, forestry research from the same US Pacific Northwest region also finds large increases in BAI in surviving conifers adjacent to silvicultural treatments[32], analogous to the edge enhancements in BAI that we report. Furthermore, a recent study on European temperate forests similarly found that temperate forest edges exhibit a 95% increase in aboveground carbon stock within 5 m of an edge[33]. Together, these results suggest that the pattern of elevated growth along forest edges holds true across large portions of the temperate forest biome.

The implications of these findings on global estimates of tree growth and carbon storage are proportional to the amount of fragmentation within temperate and tropical forest biomes. We quantified forest fragmentation throughout both types of forests using a 30-meter resolution, global, forest-cover map[29,34] (Fig. 4).

study region. The proportion of forest area within 30 m of an edge varies across ecoregions from <5 to 68% of all forest area, with an area-weighted average of 18.5% (Fig. 3a). We quantified the expected difference between interior and edge forest based on ecoregion-specific forest composition (Fig. 3b) and abiotic predictors, then combined the proportion of forest within 30 m of an

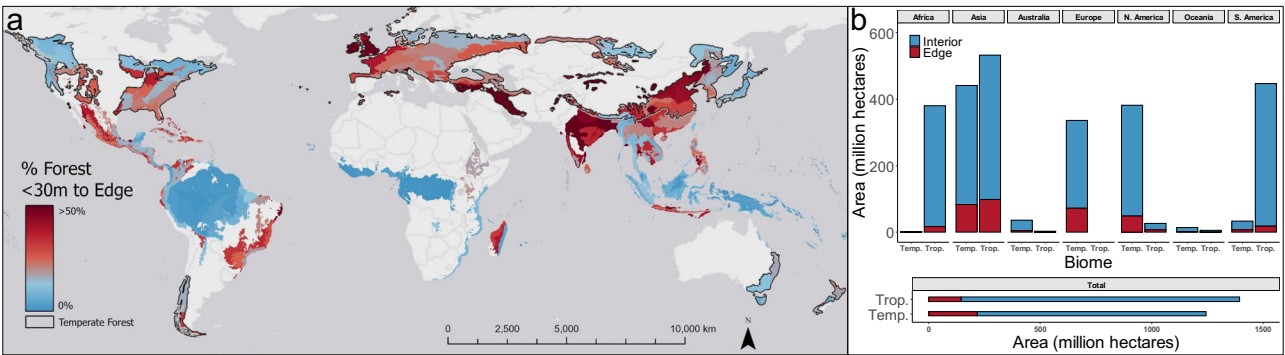

**Fig. 4 Temperate forests are nearly 1.5 times more fragmented than tropical forests. a** The percent of temperate and tropical forest area within 30 m of an anthropogenic edge within each global ecoregion. **b** The area (in millions of hectares) of edge and interior forest, grouped by biome and continent.

Temperate forests have >50% more forest area within 30 m of a forest edge than tropical forests (217 million ha compared to 143 million ha, respectively) (Fig. 4b). Europe has the highest percent of edge temperate forests (21.5%), while North America has the highest percent of edge tropical forests (29.1%) (Fig. 4a). Fragmented forests are often perceived as degraded remnants. However, the prevalence of temperate forest edges and their distinctive ecosystem functions, demonstrated here, argue for a reassessment of forest edges and fragments. These are the forests that people interact with most, they are distinct from interior forests in ways that need to be better understood, and, in some functions, are of disproportional value. The large increases in growth near forest edges that we observe here have major implications for understanding how these ecosystems will respond to ongoing fragmentation and climate change.

Emphatically, this research does not argue for proactive forest fragmentation as a prescription to increase carbon sequestration. The increased carbon storage along the edges of fragmented remnants does not come close to offsetting the loss of terrestrial carbon stocks and future sequestration capacity associated with forest loss[15]. Furthermore, there is evidence that the temperate edge responses are hindered by extreme heat, suggesting that rising global temperatures may exacerbate heat stress at temperate forest edges[15] and cause them to respond more similarly to tropical forest edges. Instead, this is a call to acknowledge the complexity of interactions between global change drivers across diverse ecosystems. Centuries of fragmentation have created a permanent shift in the microenvironment of a large and growing proportion of the global temperate forest area. With rising populations, expanding urban and agricultural areas, and ongoing deforestation, the critical need to understand fragmented forests as distinct ecosystems only grows. Any attempt to predict future forests must account for ongoing changes in the prevalence of forest edges and the potential contributions of fragments to terrestrial carbon storage.

## Methods

**Overview**. We used data from the national forest inventory conducted by the US Department of Agriculture, Forest Service, Forest Inventory and Analysis (FIA) program to quantify tree biomass and growth along forest edges and within the forest interior. We estimated the causal impact of the forest edge environment on patterns of tree biomass and growth, while accounting for potentially confounding variables. We then used the regression models to estimate the aggregate difference in growth attributable to forest edges throughout the northeastern U.S. Finally, to better understand the implications of our findings, we quantified the degree of forest fragmentation throughout temperate and tropical forest biomes world-wide, using a 30 m forest cover map.

**Study area**. Our analyses of edge impacts on forest biomass and growth were conducted throughout twenty-states (1.7 million km²) in the northeastern and upper mid-west of the United States (Supplementary Fig. 1). This region contains 765,000 km² of forest and encompasses gradients of dominant land-uses, climatic conditions, and forest composition while remaining within deciduous, coniferous, and mixed temperate forest ecosystems.

**Identifying edges in forest inventory data**. The FIA collects measurements of tree size, growth, and land-use within a nested plot design across the country[19]. Each FIA plot is composed of four individual subplots; within each subplot, the diameter at breast height (dbh) of every tree >12.7 cm is measured during each measurement period. The re-measurement frequency for FIA plots in our study area is between 5 and 7 years, but this can differ between Forest Service regions. In addition to tree measurements, the database details land-use condition data that includes the proportion of the area that is forested and, on some plots, the land-cover class of the non-forest area (FIA User's Manual, Condition Table). FIA plots are considered forested if some portion of the plot includes a contiguous forest patch (including potentially outside of the plot area) of greater than 4047 m² that has more than 10% canopy cover. With a memorandum of understanding between the USFS and Harvard University, we had access to the true, unfuzzed plot coordinates, which are not publicly available. Evaluating >48,000 plots in the USFS Northern Region sampled from 2010 to 2020 and selecting the most recent measurement cycle for each plot, we identified subplots that contained both a forest and a non-forest condition and categorized these as edges (Supplementary Table 1). Only subplots that included a forest condition in both the most recent and previous measurement were included. Subplots where the mapped condition changed from forest to non-forest were excluded. Changes in the amount of mapped forest condition were included and are incorporated into the calculation of response variables using the most recent condition area. We identified FIA plots where all four subplots were fully forested as interior plots to be used for comparison. Subplots located within the same plot as an edge subplot (i.e., edge-proximate subplots) were excluded from this study due to limitations in our ability to quantify their distance from an edge. The spatial configuration of subplots is such that a fully forested subplot may be up to ~65 m away from an identified forest edge within another subplot. Studies suggest that the distance of edge influence in temperate forest does not extend more than 30 m into the forest interior[15,33]. Since the FIA does not contain information about the geometry of non-forest conditions beyond the subplot boundary, we deemed that the large uncertainty in the relationship between these subplots to a non-forest edge precluded their inclusion in the study. The FIA plot configuration prevented quantification of the distance of edge influence in our analysis; the exclusion of subplots adjacent to edge-subplots may limit direct comparisons with other fragmentation studies.

We used the FIA condition data to characterize the non-forest land use in edge subplots. Information on adjacent non-forest land cover is not collected on all FIA plots (4327 of 6607 edge subplots). We aggregated FIA land-cover classification to a binary anthropogenic or unknown edge type designation and present results from all edge subplots and the anthropogenic edge subset (FIA User's Manual Condition Table, Section 2.4.50).

For each subplot (168 m² in area), we calculated two primary response variables of interest: total live tree BA and BAI. Notably, trees smaller than <12.7 cm diameter are only recorded within a small portion of the plot, called the microplot. Our study design prevented the inclusion of the microplot and therefore excludes trees beneath this diameter threshold. Trees that grew into the measurement size class between the previous and most recent measurement are included. The exclusion of small trees and saplings may result in a conservative estimate of the difference between edge and interior BA and BAI, as other studies have found a higher density of small-stemmed woody vegetation along forest edges[35]. BA is calculated from a single plot measurement, as the summed BA of all live adult trees (>12.7 cm dbh) in m². BAI was calculated on a per-tree basis as the difference in radial growth of live adult trees between the most recent and previous measurements, and then divided by the number of years between measurements (m² yr⁻¹). In addition, we aggregated individual tree diameter measurements to

calculate mean stem density (stems ha$^{-1}$) and mean tree diameter for each subplot (Fig. 2).

We accounted for variable subplot area by normalizing both BA and BAI to a per-hectare of forested area basis, resulting in units of m$^2$ ha$^{-1}$ and m$^2$ ha$^{-1}$ yr$^{-1}$, respectively. To account for potential small-area bias, we performed a sensitivity analysis on the relationship between BA and subplot forested area (Supplementary Fig. 2). We subsequently excluded 1284 subplots under 30 m$^2$ in area as the area to BA relationship asymptotes relationship above this threshold. Finally, we accounted for errors in field dbh measurements, sometimes resulting in negative BAI values, by excluding the <2.5% and >97.5% quantiles of both BA and BAI distributions.

Given their spatial configuration, FIA subplots are not fully independent measurements, potentially introducing issues with pseudo-replication and spatial autocorrelation within our dataset. To test for spatial autocorrelation we examined the semivariance of model residuals[36], and found that there was high correlation only at distances of less than 1 km. The spatial stratification of the FIA plot design minimizes issues of plot–plot proximity within our study. However, to account for autocorrelation between subplots, we filtered our pre-matched dataset to only including one subplot from each FIA plot. For plots containing multiple edge subplots, we selected the subplot with the largest forested area. For interior plots, we selected the central subplot and excluded all others.

### Isolating the effect of edges on growth

*Abiotic controls.* To account for environmental controls on forest growth we included the most critical abiotic predictors of terrestrial vegetation productivity (light, water, temperature, and nitrogen deposition) as covariates in the regression models (Supplementary Fig. 4, Supplementary Table 2). Light, water, and temperature data were drawn from spatial raster maps (0.5° resolution) as unit-less indices of relative limitation on vegetation productivity, ranging from 0 to 1[3]. Nitrogen data were drawn from the 2018 NADP gridded inorganic wet nitrogen deposition product (4 km spatial resolution; kg of N ha$^{-1}$)[37]. To interpolate across small gaps in the raster data (usually along water bodies), we used the Nibble tool from ArcGis Pro (ESRI Team). We then used FIA plot locations to extract values from each raster layer for all FIA subplots.

*Forest composition.* Tree species may vary in their responses to biogeochemical changes that occur on forest edges. Overall forest community response emerges from complex interactions between species. We used aggregations of tree species, termed forest composition groups (or forest types)[38], to assess if species composition influenced the response to altered edge condition. Forest type classifications for each subplot are provided by the FIA (FIA User's Manual, Condition Table) and are defined in Appendix D therein. We aggregated the FIA forest types into eight broader species groups, following Thompson et al.[23], and defined in Supplementary Table 1.

### Matching, GLM regressions, and model selection.
All statistical analyses and most of the data processing were conducted in R, version 3.4[39]. Using a causal inference framework, we created a quasi-experimental statistical design that included pre-matching followed by a GLM regression analysis[40]. Matching emulates an experimental design using observational data by identifying control groups of untreated (forest interior) plots that were as similar as possible to treated (forest edge) plots in terms of observable confounders. By capturing key differences in abiotic variables we control for the fundamental drivers of forest productivity, allowing for a direct estimation of the average treatment effect of edges. Similarity was defined by nearest-neighbor covariate matching determined by Malahanobis distance, implemented in the MatchIt library in R[41], the simplest and best method when the dataset is robust enough to find a match for every treated plot[20]. This method excludes forest interior plots that are not matched with an edge plot. Given differences in sample size between the full edge dataset and the subset designated as anthropogenic edges, we performed matching separately on the two datasets. To assess the efficacy of matching on reducing the differences in covariate distributions, we used summary statistics calculated with the MatchIt library and report the pre- and post-matched covariate balance in Supplementary Table 4 and Supplementary Table 5 (sensu Schleicher et al.[42]). Matching was highly successful, largely eliminating differences in all covariate distributions in both datasets.

Our primary response variables of interest, BA and BAI, were right-skewed, non-normally distributed and violated the assumptions of normality necessary for ordinary least squares regression[43]. We, therefore, used a GLM to better fit the structure of our data. GLMs are an extension of linear regression that allow more freedom in the choice of probability distribution function through the use of a link function to model relationships between predictors and response variables[44]. The gamma probability distribution is frequently chosen to model BA, given its assumptions of positive, continuous values and flexible model form[23,45]. We performed a series of GLM regressions on our post-matched dataset, using a gamma probability distribution with an inverse link function to model the relationship of BA and BA with a suite of predictor variables, using the *glm* function as implemented in the R Core *stats* package[39]. Due to differences in sample size between the all-edge dataset and the anthropogenic-edge subset, we modeled these two datasets separately for each of BA and BAI, resulting in four separate regression analyses. We used a model selection framework to identify the

most parsimonious model within each of the model sets based on the Akaike Information Criterion (AIC) and residual deviance statistic[46,47]. We report the model-selection and model-fit results for each of our separate analyses, including model forms, AIC, Nagelkerke Pseudo-$R^2$, and residual deviance in Supplementary Table 2. Across all four regression analyses, the best-performing model was one that included an interaction between the edge-status and forest type categorical variables, as well as the variables of temperature-limitation, light-limitation, water-limitation, and nitrogen deposition.

We then used the best performing model from each analysis to compare the differences in BA and BAI between forest edge and interior across each forest type. We estimated the treatment effect of edge-state within each forest type using the *ggeffects* package[48] to calculate marginal effects with the continuous predictors (temperature, light, water, and nitrogen deposition) held at their within-forest type regional means. The results of this analysis are displayed in Fig. 1 and Supplementary Table 3; primary error bars on the interior point show the 95% confidence interval of the marginal effect from the full edge model, while secondary error bars show the CI from the anthropogenic edge model. Due to the smaller sample size in the anthropogenic model, estimates of the mean marginal effect of the interior plots vary slightly (though non-significantly) from those from the full dataset. The main text description reports outputs from both models, calculated from separate interior mean estimates. For visual clarity, we only display one set of interior means in Fig. 1.

*Mortality and timber harvest.* In tropical forests, large reductions in productivity along edges are associated with increased tree mortality.[9] To assess differences in tree mortality across our study region, we applied a simplified GLM analysis, including edge-state as our only predictor variable. The FIA differentiates between mortality attributed to timber harvest and that attributed to other, non-harvest causes. The results of this analysis are presented as marginal effects of each edge category in Supplementary Fig. 3. There are no significant differences in biogenic mortality between edge groups and no difference in overall mortality (combined biogenic and anthropogenic); there is a small, but statistically significant ($p < 0.001$), increase in harvested BA within both all-edge and anthropogenic edges as compared with the forest interior. We note that the exclusion of small-diameter trees from our study could alter these results if there was differential mortality between edge and interior in smaller tree size classes.

Temperate forests are heavily impacted by forest management[49]. We tested the robustness of the effect of edges on growth and biomass by withholding all subplots with a record of anthropogenic removals on the full FIA plot (i.e., management; $n = 3642$). We found no difference in the overall effect of edges nor meaningful differences within forest type groups.

### Scaling edge effects on forest growth across the Northeast.
Ecoregions are a widely used geographic partitioning of ecosystems into coherent spatial units as defined by abiotic, biotic, and anthropogenic characteristics[28]. EPA Level IV ecoregions are delineated by differences in environmental characteristics analogous to those that we used to model forest growth and thus are a comparable spatial unit to quantify the aggregated effects of fragmentation.

*Quantifying fragmentation.* To quantify anthropogenic forest edge area, we identify forest cover within 30 m of a road, development, or agricultural field (sensu Smith et al.[6]) using a 30 m resolution land-cover product from 2016 of the National Land Cover Database (NLCD)[50]. Edge forest was defined as all forest pixels adjacent (queen's rule) to a non-forest cultivated or developed pixel (Supplementary Fig. 6a). Figure 3a shows the percentage of total forest area classified as edge within each ecoregion. We report that 18% of the total forest area in our study domain is adjacent to an anthropogenic edge. Differences from the reported 22% in Smith et al. are likely attributable to the use of a different NLCD product. Note that the definition of forest edge here may differ from that of the FIA analysis, given the constraints on quantification of the distance of edge influence and the spatial resolution of the land cover products.

*Ecoregion edge impacts.* To scale the effects as illustrated in Fig. 3, we quantified ecoregion forest composition by (1) using a 250 m resolution USFS forest type map[27], we aggregated raw forest type values to the aggregated forest type groups included in our regression models (Figs. 3b), (2) we calculated the total area of each forest type group within each ecoregion, then used the average temperature, light, water, and nitrogen deposition in each ecoregion as inputs to our GLM regression models to calculate the BAI of edge and interior forest for each forest type. With the proportional area of each forest type, we calculated an area-weighted mean and then differenced the estimated edge and interior BAI to produce an expected difference of forest growth (BAI m$^2$ ha$^{-1}$) between edge and interior within each ecoregion (Supplementary Fig. 5). Finally, we combined the proportion of edge forest with the expected growth difference to quantify the estimated difference in percent increases in ecoregion BAI within each ecoregion attributable to increases of forest growth at the edge (Fig. 3c).

### Quantifying global forest fragmentation.
We quantified the extent of forest fragmentation throughout temperate and tropical forests worldwide at the scale of

ecoregions using the Hansen Global Forest Change (v1.7)[51] dataset on Google Earth Engine (GEE)[52]. Tropical and temperate biomes were delineated in a global ecoregion map[53], analogous to the more detailed ecoregions described earlier. The tree canopy cover layer from the Hansen dataset provided estimates of percent tree canopy cover for the year 2000 at 30 m resolution globally produced by time series analysis of Landsat images[51]. To calculate the percentage of edge forest in each ecoregion: (1) a 10% threshold (following the FIA definition of minimum forest cover[19]) was applied to the tree canopy cover layer to separate forest and non-forest pixels, (2) each forest pixel adjacent (queen's rule) to a non-forest pixel was classified as edge forest on GEE, and (3) ArcGIS Zonal Statistics Tool was used to calculate the percentage of edge forest in each ecoregion. Definitions of forest cover via % canopy cover vary between studies, therefore we performed a robustness check on our results to the threshold definition of forest cover by re-analyzing with a 30% canopy threshold. While there were differences in the calculated raw area of forest edges, the ratio of area fragmented between temperate and tropical forests did not change meaningfully (Supplementary Fig. 7). We then compared the Hansen-derived forest fragmentation to the 2016 NLCD-derived forest fragmentation used in our previous analysis to assess comparability of the two products. Supplementary Fig. 8 shows the agreement between the percent edge forest values calculated based on the two forest maps for the 247 ecoregions in the Northeast US. The agreement is strong especially in large and more forested ecoregions. The Hansen-derived percent edge forest explained 84.5% of the variance in NLCD-derived percent edge forest with RMSE of 6.1 (%) at ecoregion level. The spatial aggregation to ecoregion level largely reduced the uncertainty in the mapping of forest pixels in both products.

**Reporting summary**. Further information on research design is available in the Nature Research Reporting Summary linked to this article.

## Data availability

The processed, post-matching FIA data generated in this study and used to generate Figs. 1 and 2 have been deposited in the Harvard Forest Data Archive under accession code HF419[54]. The spatially aggregated estimates of BAI presented in Fig. 3c and summaries of global forest edge area displayed in Fig. 4a are available in the HF Data Archive under accession code HF419. The un-fuzzed FIA location data are protected and are not available due to data privacy laws. Unprocessed FIA inventory data is available at https://apps.fs.usda.gov/fia/datamart/. The National Land Cover Database land cover layer is at https://www.mrlc.gov/data. The forest cover map we use for the global analysis is available on Google Earth Engine.

## Code availability

Statistical analyses and FIA data-processing were conducted in the R programming environment, version 3.4. Generalized linear model regressions were performed using the R *stats* package, version 3.4.3. Marginal effect estimates were calculated using R package *ggeffects*, version 1.1.1. Other GIS analyses were performed in the ESRI software ArcGIS Pro, version 2.4. The global analysis of forest fragmentation was performed in Google Earth Engine. The code used to analyze and process FIA data are not available publicly due to data privacy laws.

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

## Acknowledgements

We thank the many colleagues who gave us friendly feedback throughout this research, in particular to C. Canham, S.C. Wofsy and D. Foster for their thoughtful suggestions, J. Holt and A. Kalinin for their statistical guidance. FIA plot location data was made available via Memorandum of Understanding 09MU11242305123 between the U.S. Forest Service and Harvard University. Funding: This work was supported, in part, by the United States Department of Agriculture National Institute of Food and Agriculture Award 2017-67003-26487, the Harvard Forest LTER Program (NSF DEB 18-32210), the Rafiki B. Hariri Institute at Boston University and by a National Science Foundation Research Traineeship (NRT) grant to Boston University (DGE 1735087).

## Author contributions

L.M., J.T., L.H., and A.R. conceived the project and designed the study. L.M. processed the FIA data and performed the subsequent analyses. X.T. and L.M. performed the global edge analysis. All authors contributed to the writing and intellectual development, and gave feedback throughout the project.

## Competing interests

The authors declare no competing interests.

## Additional information

**Peer Review Information** *Nature Communications* thanks Rico Fischer and the other, anonymous, reviewer(s) for their contribution to the peer review of this work. Peer reviewer reports are available.

