## [Peer Review File · Nature Communications]

Reviewer comments, first round review –

Reviewer #1 (Remarks to the Author):

The authors investigated the impacts of fragmentation on the growth of temperate forests in the USA, especially on the edge area of these forests. They studied the differences of basal area, forest growth, and mortality between forest edge and forest interior. For this purpose, they analyzed 48,000 forest plots in the northeast of the USA between 2010 and 2017. They have found that basal area and basal area increment is higher in the edge area compared to forest interior, but see no differences in tree mortality. These results differ from study results in tropical forests. The study also explained what the drivers (e.g. sunlight) are and why it is different in the tropics.

The authors have done an impressive work in analyzing a large dataset of empirical data and combining this with remote sensing maps. The manuscript is well written and the figures are well designed. However, I have found weaknesses in the statistical methods and furthermore, the authors have ignored significant driving factors such as forest age and forest management.

Major points:

1. The title and the abstract (and L41-43) are somehow misleading in several aspects: forests in the Northeastern United States were investigated and not all regions with temperate forests. Whether the results found can be transferred to other temperature forests remains open. For example, the authors themselves note that the forests in NW-USA probably behave more like tropical forests (L125-127). Is it possible to use the full inventory dataset of the US and extent the analysis? In addition, the basal area and basal area increment were studied here, not biomass nor forest productivity. These facts should be presented more clearly. Moreover, the claim that temperate forests are twice as fragmented as tropical forests is unclear.
2. The statistical model used is poorly described. There is no equation for the model (only a strange type of equation after FigS3) and it is unclear what the variables mean (e.g. the units are missing). Furthermore, in my opinion important variables were not taken into account for the statistical model for predicting basal area and basal area increment. The basal area and its growth depends very much on the age of the forest. Also the age of edge creation is important for differences between edge/interior. An even bigger influence has the forest management, which was ignored here. Almost all temperate forests are somehow managed, with strong implications on basal area and growth. Forest age and management should be considered in the basal area estimation model. In addition, the goodness-of-fit of the statistical model was not described. Information on R², RMSE, bias etc. are missing. So I cannot judge whether the statistical model can reproduce the basal area and its increment.
3. There are not many results in this study (only Fig1 and Fig2c). Here I would like to see more in-depth analyses. I also find the results on mortality very exciting and would recommend to show them in the main text as well. Furthermore, the influence of the cardinal direction could be another important point, as it strongly influences the tree growth at the edge of the forest (position of the sun).
4. In my opinion, the comparison with the tropics (see Fig 3) was not done properly. The presentation of the data in Fig 3 has two weaknesses: the relative amount is strongly dependent on

the size of the ecoregion. For example, the Amazon is much larger than a very small ecoregion in NE-USA, resulting in a large relative amount of edge area in the US (shown in red) - but the absolute amount of edge area is much higher in the Amazon. Furthermore, a very low value of 10% forest cover was used as threshold for forest area definition (actually 30%-70% is used in other studies). This leads to the fact that in tropical areas there is supposedly a lot of connected forest even in regions with high deforestation. The authors should reconsider this analysis or leave it out, since the study has enough interesting results for forests in the US.

5. In a quick internet search I found similar results for temperate forests in Europe (see e.g. Meeussen et al. "Drivers of carbon stocks in forest edges across Europe" 2020 Science of the total env.). In this study they already found higher values in the forest edge area. This result should also be compared and discussed to get a larger picture.

Detailed comments:

-Abstract L18-L20: In the manuscript mainly the basal area and basal area increment is examined. I find it very confusing that the abstract is about biomass and forest productivity. Please be more precise.

-L22: I do not see the difference in the fragmentation between tropical forests and temperate forests. Here the relative edge area fraction was investigated. Please be more precise.

-L44: Please mention the typical area of such a FIA plot, which is with <200m² rather small with a lot of uncertainty.

-L59: I really like the distinction between natural edges and anthropogenically formed edges. This is a great added value for this study.

-Fig1: Very interesting figure. I miss the presentation of the total basal area and the basal area increment over all forest types in this figure. Although the numbers are mentioned in the text, this would be a very helpful addition.

-Fig1: I wonder if the basal area increment in the anthropogenic edges is much higher because the total basal area in the edge is also higher. So I recommend to analyze the relative basal area increment (BAI/BA). Are there still differences between edge and interior if investigating relative increment?

-From my point of view, the study of mortality is a very important result because it was most surprising for me and it shows most clearly the difference to the tropics. Perhaps the authors could consider including Fig S4a in the main text.

-L81: "When isolating anthropogenic edges..." How is this done?

-L96: The calculation of the relative number "18.5%" is unclear. Is this the mean value over all relative numbers (5% to 68%), is the mean value weighted by the size of the ecoregion?

-Fig2c: The figure is somewhat misleading, since the relative growth is dependent on the total forest area, isn't it? One could also represent this map with absolute values of basal area increment and report from this the total basal area increment due to edge effects.

-References: Some references are incomplete. E.g. for ref 18 the publication year is missing

Reviewer #2 (Remarks to the Author):

The manuscript titled Fragmentation impacts on temperate forest productivity: reversal of the tropical edge paradigm presents estimates of basal area and basal area increment for forest and forest edges in the northeastern US. The paper is well written, organized, and the results support the conclusions. This is an interesting and valuable use of the US national forest inventory data and the methods to extend the analysis to temperate fragmented landscapes more broadly provides context and highlights the importance of edges in terms of forest ecosystem structure and function and differences with tropical forest edges. That said, there are a few areas in the manuscript that require clarification and potentially reanalysis. First, and most importantly, the way in which forest edge was spatially defined using the FIA subplots may differ from how it is defined using the land cover data product to assess edge within and beyond the study domain. While this may have little influence on the overall findings of the paper, consistency between approaches is important and may have implications on the estimates of forest growth and biomass reported. This issue may be resolved with clarification on the methods used or by analyzing the FIA data at a different spatial scale to be consistent between approaches. Second, while basal area and BAI are used throughout, it would be useful to also include diameter distributions by edge category to better illustrate potential differences in stand structure that may be masked by comparisons of basal area alone. This may also provide support for the patterns in BAI observed across edge categories. Next, the way in which the analysis was conducted (presumably) prevented the incorporation of micro-plot estimates of small diameter trees from being included in the study. Given the importance of these trees in estimates of BA and BAI, some discussion of why they were not included is warranted as well as how ingrowth trees were handled between remeasurements. Finally, more details on the subplots that were included and those that were not included are needed, particularly for estimating BAI from remeasurements. It is not clearly why, in a region with multiple complete remeasurements, why there were so few remeasured subplots available for this analysis. Details on each of these topics among others are listed below.

L44: Given this is an international journal it would be helpful to further describe the FIA program. You may consider describing in as “the national forest inventory (NFI) conducted by the United States (US) Department of Agriculture Forest Service Forest Inventory and Analysis (FIA) program.”

L46: FIA remeasurements in the eastern US are on a 7-year cycle and some states “buy down” to a 5-year (quinquennial) cycle. Consider describing remeasurements in the study domain as occurring every “5-7 years”. It may also be worth mentioning here (rather than in the methods) given how the measurements are being used in the study that the FIA plots are fixed area and permanent.

L56: It may be worth noting how forest land is defined by the FIA program so that the reader has a clear understanding of how “interior” forest is defined in the context of the forest land definition as well as how edges were defined.

L59-67: Did you consider looking at annual net growth or net change over the remeasurement

periods included in the study which specifically accounts for mortality and net loss because of damage, rot, broken top, or other causes. The net change estimates further include harvest removals.

L66 and elsewhere: The results reported here and elsewhere suggest statistical tests were conducted to assess differences in categories. Where noting significant and non-significant results it would be helpful to report statistics and reference the methods section where the tests should be described.

Figure 1: The 95% confidence intervals reflect, to some extent, the sample sizes for the three categories by forest type group. A table describing the sample sizes for the three groups would be helpful to understand the distribution of plots falling into the different categories.

L128: This is an interesting and useful expansion of the results presented in the paper. That said, the way in which the assessment of edge was conducted may have implications on how the FIA subplot analysis was conducted and how plots were included and excluded – see below for more details.

Methods

Study area: “twenty-state” should be “twenty-states”

Fig. S1: I assume that approximate plot locations (the perturbed locations) were used to construct this figure? If not, consult with your MOU FIA partner. If so, consider including the “approximate locations” in the figure capture so that readers understand you are not sharing actual plot locations.

Identifying edges in forest inventory data

As before, plots in the eastern US are on a 7-year measurement cycle with some states “buying down” to a 5-year cycle.

Why were subplots conditions selected rather than conditions on plots? While the subplots are more specific, given the spatial arrangement, the subplots are correlated and treating them independently without considering this correlation may lead to misleading results. While it would greatly reduce your subplot sample size, using only the central subplot would alleviate the necessity to account for correlation between subplots and would also provide the most accurate geo-location since a GPS location (rec grade GPS receivers with accuracies ± 10 meters) is only collected at the central subplot (subplot 1) and distance and azimuth are used to estimate subplot locations for the other 3 subplots.

Further, given the proximity of the subplots, there may be an edge effect to fully forested subplots adjacent to subplots which are partially forested (multiple subplot conditions forest and non-forest) or non-forest. Such cases, which are common in the FIA data (if only plots with at least one forest land condition were included in the analysis then most of the 29,175 subplots likely fall into this category), would be similar to the way you have quantified the area that is fragmented in your global land cover analysis. This same logic would hold for plots that are fully forested but may be adjacent to a forest edge that is not captured on plot. It may be worth noting this possibility in the manuscript.

How were subplots where the mapped conditions changed between measurements treated?

While only trees ≥ 12.7 cm in diameter were included in the analysis, how were ingrowth trees, those they passed the diameter threshold between measurements treated in terms of BA and BAI estimates?

While I appreciate that small diameter trees were not considered in the analysis based on the way edge was determined on subplots (micro plots may not have encompassed both forest and non-forest conditions) it may be worth noting the role of small diameter trees in both the BA and BAI estimates in the paper. Again, if you conducted at the plot level you could consider the small diameter trees but if you restricted the analysis to the central subplot you could not. Basal area can be missing leading in terms of characterizing forest conditions, particularly forest structure. Including diameter distributions as a figure across the three edge-non-edge categories would help the reader understand if stand structure was markedly different between the categories. This would also lend support for the BAI estimates across categories – are edge forests smaller diameter (suggesting younger) resulting in higher BAI than “interior” forest which may be the same BA with fewer large diameter (older) trees but with very different BAI?

Given the study period and the remeasurement lengths for eastern FIA plots how did less than half of the forest-non-forest subplot conditions not have remeasurements? Was it because the map conditions changed between measurements? This needs further clarification to understand the differences. A table of n subplots by category would be helpful to understand samples sizes used in the regression analysis and would help, in part, to explain the CI in Figure 1 – sample size is likely a factor (following up on this as later in the methods this is acknowledged).

Matching and linear regressions

This is an interesting analysis and use of the NFI data. That said, some of the points I raised above may influence the matching of edge and interior plots.

Mortality and timber harvest

As before, how might including small diameter trees < 12.7 cm dbh contribute to these results. Also, the FIA program provides estimates of net annual growth (which includes mortality and damage) and net change (which includes mortality, damage, and harvest removals) based on individuals trees which might be considered as an alternative to BAI for estimates of biomass growth or carbon sequestration.

Reviewer #3 (Remarks to the Author):

This manuscript characterises the impacts of edge effects on forest structure and growth in North Eastern USA, using data on a national forest inventory. The authors use a linear model to analyse the data and make predictions about the impacts of forest edges across the region. The authors conclude that forest edges enhance both basal area and basal area and that this varies across different forest types. In addition, anthropogenic edges resulted in larger enhancements of basal area and basal area increment. Using these models to scale up to a regional level, the authors concluded that edges resulted in $\sim 6\%$ increase in BAI across forest in the North Eastern USA. This

paper presents a novel result and is generally well written. However, I do have some doubts about the robustness of the methods used by the authors that I would like them to address.

Please note that my comments focus mainly on the problems that I perceive with the paper. This is simply due to my time constraints and not a reflection on the work. I try to focus on what I think will help to improve the paper most, but spend less time mentioning all the things that are great about the paper.

Major issues

- In the section on statistical matching, the authors state that ‘Matching successfully eliminated differences in all covariate distributions in both datasets.’ However, in their review of matching methods Schleicher et al (2020) state that ‘...in most instances matching reduces—but does not eliminate—differences between treatment and control units.’ As such, I would like to see evidence of how much difference there was between covariates used for matching both prior to and after statistical matching was undertaken. Schleicher et al (2017) provide a potential way in which to present this, which you can find in their supplementary materials.
- In the materials and methods section, the authors describe using a linear regression model to estimate differences in basal area (BA) or basal area increment (BAI) between edge and core forest areas. Did the authors do any model simplification as part of this? The aim of any linear modelling should be to produce the simplest model that provides a good fit for the data. Without this, it isn’t clear that edge and forest type are important predictors of BA or BAI in the current study. The authors should redo this analysis and include some form of model selection/simplification.
- Regarding the modelling of BA and BAI, it would also be a good idea to examine any potential spatial autocorrelation that may be driving apparent relationships. At the very least, the authors should produce a figure to show how the residuals of their model vary in space.

Minor issues

- Line 46 – Best not to use the word ‘quinquennial’ here as nobody will know what it means, better to say ‘...collects measurements every five years...’ instead.
- Line 48 – BA is not exactly an analog of biomass since biomass calculations usually incorporate height as well. I usually just describe BA as a measure of forest structure.
- Figure 1 – I’m not clear why, for the model that examines the effects of anthropogenic effects, you don’t also have a result for edges that were ‘natural.’ From the methods section, it seems that you classified edges into these two groups so I would expect to see the results for these groups reported here.
- Lines 84-89 – Say which regions this relates to. As someone who isn’t from North America this wasn’t clear to me.
- Lines 129-130 – The authors say that they quantified fragmentation in both tropical and temperate forest biomes, referring to Figure S6. However, Figure S6 is a comparison of fragmentation between different remote sensing products for the North Eastern USA. The authors should correct this.
- Figure 3 – It is quite hard to make out where temperate regions are on this map as the line is quite thin. Maybe this data would benefit from being a composite figure with a plot next to it showing the mean/median fragmentation of each ecoregion as well as the range of values in temperate vs tropical regions.
- In the methods section (Scaling edge effects on forest growth across the Northeast) the authors detail aggregation of the potential effects of edges on forest growth at the ecoregion level. What

was the rationale for doing this at the ecoregion scale instead of aggregating using a finer scale grid? I worry that this aggregation may hide some of the inevitable within ecoregion variation. Similarly, I'm not clear why temperature, light, water, and nitrogen deposition were averaged across ecoregions as it is clear from figure S3 that you have relatively high resolution data on this.

References

Reviewer #1 (Remarks to the Author):

The authors investigated the impacts of fragmentation on the growth of temperate forests in the USA, especially on the edge area of these forests. They studied the differences of basal area, forest growth, and mortality between forest edge and forest interior. For this purpose, they analyzed 48,000 forest plots in the northeast of the USA between 2010 and 2017. They have found that basal area and basal area increment is higher in the edge area compared to forest interior, but see no differences in tree mortality. These results differ from study results in tropical forests. The study also explained what the drivers (e.g. sunlight) are and why it is different in the tropics.

The authors have done an impressive work in analyzing a large dataset of empirical data and combining this with remote sensing maps. The manuscript is well written and the figures are well designed. However, I have found weaknesses in the statistical methods and furthermore, the authors have ignored significant driving factors such as forest age and forest management.

Major points:

1. The title and the abstract (and L41-43) are somehow misleading in several aspects: forests in the Northeastern United States were investigated and not all regions with temperate forests.

We appreciate this suggestion and have revised the title to “Elevated growth and biomass along temperate forest edges”. We also updated language in the abstract and lines 42-52, to be more reflective of our study. Lines 175-183 include a discussion of other temperate studies, putting these results into a broader context.

Whether the results found can be transferred to other temperature forests remains open. For example, the authors themselves note that the forests in NW-USA probably behave more like tropical forests (L125-127). Is it possible to use the full inventory dataset of the US and extent the analysis?

Unfortunately, our study area is primarily limited by our MOU with the US Forest Service that provides us access to the true FIA plot coordinates for only this 20-state region. Furthermore, moving outside of this study region would necessitate inclusion of distinct disturbance regimes, particularly fire, which would require additional analyses that are beyond the scope of this study.

We have added more text (L160-183) to the discussion to address the transferability of our results.

In addition, the basal area and basal area increment were studied here, not biomass nor forest productivity. These facts should be presented more clearly.

We have clarified that the primary response variables of this study are basal area and basal area increment (L46-48), correlates of biomass and growth, and we changed language throughout the paper to reflect this.

Moreover, the claim that temperate forests are twice as fragmented as tropical forests is unclear.

We substantially expanded our analysis of temperate & tropical forest fragmentation and quantified the difference in the area of fragmentation. We have modified the text and figures to reflect the updated values (L184 – 190; Figure 4).

2. The statistical model used is poorly described. There is no equation for the model (only a strange type of equation after FigS3) and it is unclear what the variables mean (e.g. the units are missing).

We thank Reviewers #1 & #3 for their comments on our statistical methodology and have substantially expanded our model analyses and reported results accordingly. We have added text to the main text and methods (L64-66; Methods Section: Matching, GLM regressions, and model selection) to reflect these changes, and have added tables to the SI to better document the variables and model forms.

Furthermore, in my opinion important variables were not taken into account for the statistical model for predicting basal area and basal area increment. The basal area and its growth depends very much on the age of the forest. Also the age of edge creation is important for differences between edge/interior. An even bigger influence has the forest management, which was ignored here. Almost all temperate forests are somehow managed, with strong implications on basal area and growth. Forest age and management should be considered in the basal area estimation model.

We appreciate the reviewer's comments about the importance of forest management and age of edge creation to our results. We have expanded our discussion and analyses of the effects of forest management on mortality (L100-105; Methods Section: Mortality and timber harvest).

Furthermore, we have performed a robustness test on our core results of edge-interior differences by withholding all FIA plots that had a record of forest management (L105 – 108). While we agree with the reviewer that the age of edge creation is likely important to differences between edge and interior, assessing the role of stand age is beyond the scope of this study. We are unaware of any available datasets that could reliably be used to assess tree ages in these multi-aged stands. We have expanded our discussion regarding the role of management and stand age (L160-169) and we hope that future research will explore further.

In addition, the goodness-of-fit of the statistical model was not described. Information on R², RMSE, bias etc. are missing. So I cannot judge whether the statistical model can reproduce the basal area and its increment.

We have added tables in the SI (Table S2) that report goodness-of-fit and relative model performance (Residual Deviance & AIC, respectively).

3. There are not many results in this study (only Fig1 and Fig2c). Here I would like to see more in-depth analyses. I also find the results on mortality very exciting and would recommend to show them in the main text as well. Furthermore, the influence of the cardinal direction could be another important point, as it strongly influences the tree growth at the edge of the forest (position of the sun).

We thank the reviewer for their comments on our mortality results. We have expanded our discussion of the mortality results (L100-109) as well as added additional results on the tree diameter and stem density (Figure 2). We agree that cardinal direction is a potential modifying mechanism on the effect of edges, but we are unable to assess that due to limitations of the dataset. We hope to address this question in future work.

4. In my opinion, the comparison with the tropics (see Fig 3) was not done properly. The presentation of the data in Fig 3 has two weaknesses: the relative amount is strongly dependent on the size of the ecoregion. For example, the Amazon is much larger than a very small ecoregion in NE-USA, resulting in a large relative amount of edge area in the US (shown in red) - but the absolute amount of edge area is much higher in the Amazon. Furthermore, a very low value of 10% forest cover was used as threshold for forest area definition (actually 30%-70% is used in other studies). This leads to the fact that in tropical areas there is supposedly a lot of connected forest even in regions with high deforestation. The authors should reconsider this analysis or leave it out, since the study has enough interesting results for forests in the US.

Following your suggestions, we have added a quantification of absolute area (Fig. 4b) to improve our comparison. Furthermore, we have performed a robustness test to our selection of the threshold of forest cover (using both 10% (Fig. 4) and 30% (Fig. S7)), and found that while the raw area values change, the overall pattern and conclusions are unchanged. We report the results of this test in the Methods (L404-408).

5. In a quick internet search I found similar results for temperate forests in Europe (see e.g. Meeussen et al. “Drivers of carbon stocks in forest edges across Europe” 2020 *Science of the total env.*). In this study they already found higher values in the forest edge area. This result should also be compared and discussed to get a larger picture.

Thank you for pointing this paper out; its exclusion was not intentional as it was published after we had begun the initial review process. We have included it in our discussion of the transferability of our findings.

Detailed comments:

-Abstract L18-L20: In the manuscript mainly the basal area and basal area increment is examined. I find it very confusing that the abstract is about biomass and forest productivity. Please be more precise.

Please see the earlier response, we have altered the language in the abstract and throughout the manuscript to better reflect our measured response variables.

-L22: I do not see the difference in the fragmentation between tropical forests and temperate forests. Here the relative edge area fraction was investigated. Please be more precise.

We have expanded our global fragmentation analysis and are now able to discuss differences between the two.

-L44: Please mention the typical area of such a FIA plot, which is with <200m² rather small with a lot of uncertainty.

We expanded our methodological description in the main body (L55-63) and the methods (L227-235) of the paper to address this point.

-L59: I really like the distinction between natural edges and anthropogenically formed edges. This is a great added value for this study.

Thank you very much!

-Fig1: Very interesting figure. I miss the presentation of the total basal area and the basal area increment over all forest types in this figure. Although the numbers are mentioned in the text, this would be a very helpful addition.

We have added a table to the SI that includes the marginal effects with confidence intervals, including values for differences across all forest types (Table S3).

-Fig1: I wonder if the basal area increment in the anthropogenic edges is much higher because the total basal area in the edge is also higher. So I recommend to analyze the relative basal area increment (BAI/BA). Are there still differences between edge and interior if investigating relative increment?

We tested our results with relative basal area increment as a response variable. We found that the patterns we report between edge and interior BAI remained consistent and statistically significant (significance assessed via Wald's test). Between all edges and interior we found a 14.9% relative increase in rBAI ($p < .0001$). Between anthropogenic edges and the interior forest we found a 21.1% relative increase in rBAI ($p < .0001$). Given the broad consistency, we chose to retain our original approach as the use of a ratio response variable may be harder to interpret for most readers.

-From my point of view, the study of mortality is a very important result because it was most surprising for me and it shows most clearly the difference to the tropics. Perhaps the authors could consider including Fig S4a in the main text.

We agree that the mortality results are very important. We expanded the analysis and discussion of both biogenic and harvest mortality (L97-108). Further, we have expanded the data figure (Fig. S3), but would prefer to leave the results in the SI given the lack of observed differences.

-L81: “When isolating anthropogenic edges ···” How is this done?

We use information on adjacent non-forest land covers collected by the FIA to classify an edge as anthropogenic. We have added clarifying text and expanded our methods section to clarify (L255-259).

-L96: The calculation of the relative number “18.5%” is unclear. Is this the mean value over all relative numbers (5% to 68%), is the mean value weighted by the size of the ecoregion?

We have amended the text to clarify that this is an area-weighted average.

-Fig2c: The figure is somewhat misleading, since the relative growth is dependent on the total forest area, isn't it? One could also represent this map with absolute values of basal area increment and report from this the total basal area increment due to edge effects.

We appreciate this feedback and have altered Figure 3 for clarity. We have replaced panel B with a forest map to help the reader understand how much forest area is represented in each ecoregion. Furthermore, we have added Figure S5 to show the absolute values of basal area increment attributable to anthropogenic edges, per the reviewer's suggestion.

-References: Some references are incomplete. E.g. for ref 18 the publication year is missing

We have corrected the incomplete references.

Reviewer #2 (Remarks to the Author):

The manuscript titled Fragmentation impacts on temperate forest productivity: reversal of the tropical edge paradigm presents estimates of basal area and basal area increment for forest and forest edges in the northeastern US. The paper is well written, organized, and the results support the conclusions. This is an interesting and valuable use of the US national forest inventory data and the methods to extend the analysis to temperate fragmented landscapes more broadly provides context and highlights the importance of edges in terms of forest ecosystem structure and function and differences with tropical forest edges. That said, there are a few areas in the manuscript that require clarification and potentially reanalysis.

First, and most importantly, the way in which forest edge was spatially defined using the FIA subplots may differ from how it is defined using the land cover data product to assess edge within and beyond the study domain. While this may have little influence on the overall findings of the paper, consistency between approaches is important and may have implications on the estimates of forest growth and biomass reported. This issue may be resolved with clarification on the methods used or by analyzing the FIA data at a different spatial scale to be consistent between approaches.

We thank the reviewer for their constructive suggestions. We have added text (L246-254, L379-381) to address potential implications of differences in spatial scale and expanded our methods section to clarify our definitions of forest edges throughout.

Second, while basal area and BAI are used throughout, it would be useful to also include diameter distributions by edge category to better illustrate potential differences in stand structure that may be masked by comparisons of basal area alone. This may also provide support for the patterns in BAI observed across edge categories.

We greatly thank the reviewer for this suggestion! We have added new results (Fig. 2; L84 – L96) to the main text to include both diameter distributions and stem densities.

Next, the way in which the analysis was conducted (presumably) prevented the incorporation of micro-plot estimates of small diameter trees from being included in the study. Given the importance of these trees in estimates of BA and BAI, some discussion of why they were not included is warranted as well as how ingrowth trees were handled between remeasurements.

We have expanded the methods section to include more detail about which trees are included in our analyses. We have also added discussion in the main text (L81-83) and methods (L262-267) to acknowledge the potential impact of excluding small diameter trees.

Finally, more details on the subplots that were included and those that were not included are needed, particularly for estimating BAI from remeasurements. It is not clearly why, in a region with multiple complete remeasurements, why there were so few re-measured subplots available for this analysis.

We thank the reviewer for flagging this issue for us. We have re-run our analyses with a more recent version of the FIA database (including updated coordinates). We have also corrected a mistake in the identification of edges that was causing some plots to be double-counted and making the amount of re-measurements seem less than it actually was. Together, these changes have fixed issues with the apparent lack of re-measured subplots. We have expanded the methods section to better specify our inclusion and exclusion criterion for the subplots.

Details on each of these topics among others are listed below.

Detailed comments:

L44: Given this is an international journal it would be helpful to further describe the FIA program. You may consider describing in as “the national forest inventory (NFI) conducted by the United States (US) Department of Agriculture Forest Service Forest Inventory and Analysis (FIA) program.”

We expanded our description of the FIA dataset to be more thorough and understandable to an international audience.

L46: FIA remeasurements in the eastern US are on a 7-year cycle and some states “buy down” to a 5-year (quinquennial) cycle. Consider describing remeasurements in the study domain as occurring every “5-7 years”. It may also be worth mentioning here (rather than in the methods) given how the measurements are being used in the study that the FIA plots are fixed area and permanent.

We have corrected the description of re-measurements and added to the description of the FIA database in the main text.

L56: It may be worth noting how forest land is defined by the FIA program so that the reader has a clear understanding of how “interior” forest is defined in the context of the forest land definition as well as how edges were defined.

We have included the FIA definition of forest land in the methods section (L233-25) to help the reader better understand how we defined forest edge and interior.

L59-67: Did you consider looking at annual net growth or net change over the remeasurement periods included in the study which specifically accounts for mortality and net loss because of damage, rot, broken top, or other causes. The net change estimates further include harvest removals.

We have expanded our analysis regarding mortality and better quantified removals and biogenic mortality (L100 -108; Mortality and Timber Harvest section in the Methods).

L66 and elsewhere: The results reported here and elsewhere suggest statistical tests were conducted to assess differences in categories. Where noting significant and non-significant results it would be helpful to report statistics and reference the methods section where the tests should be described.

We appreciate the suggestion and have added results of significance tests throughout the manuscript.

Figure 1: The 95% confidence intervals reflect, to some extent, the sample sizes for the three categories by forest type group. A table describing the sample sizes for the three groups would be helpful to understand the distribution of plots falling into the different categories.

We have extended Table S1 to include sample sizes for each forest type and edge category.

L128: This is an interesting and useful expansion of the results presented in the paper. That said, the way in which the assessment of edge was conducted may have implications on how the FIA subplot analysis was conducted and how plots were included and excluded – see below for more details.

Thank you for the compliment! We have responded to this point below.

Methods

Study area: “twenty-state” should be “twenty-states”

We made this correction.

Fig. S1: I assume that approximate plot locations (the perturbed locations) were used to construct this figure? If not, consult with your MOU FIA partner. If so, consider including the “approximate location” in the figure capture so that readers understand you are not sharing actual plot locations.

Thank you for catching this! We have amended the caption to say approximate locations.

Identifying edges in forest inventory data

As before, plots in the eastern US are on a 7-year measurement cycle with some states “buying down” to a 5-year cycle.

We have corrected this.

Why were subplots conditions selected rather than conditions on plots? While the subplots are more specific, given the spatial arrangement, the subplots are correlated and treating them independently without considering this correlation may lead to misleading results. While it would greatly reduce your subplot sample size, using only the central subplot would alleviate the necessity to account for correlation between subplots and would also provide the most accurate geo-location since a GPS location (rec grade GPS receivers with accuracies ± 10 meters) is only collected at the central subplot (subplot 1) and distance and azimuth are used to estimate subplot locations for the other 3 subplots.

We appreciate the reviewer’s point about the potential of pseudo-replication by using the subplots as our unit of study. To address this (along with the issue of spatial auto-correlation raised by reviewer #3), we have altered our analyses to only include one subplot from within a single FIA plot. We describe this process in our methods section (L280-287).

We describe this process in our methods section (L280-287).

Further, given the proximity of the subplots, there may be an edge effect to fully forested subplots adjacent to subplots which are partially forested (multiple subplot conditions forest and non-forest) or non-forest. Such cases, which are common in the FIA data (if only plots with at least one forest land condition were included in the analysis then most of the 29,175 subplots likely fall into this category), would be similar to the way you have quantified the area that is fragmented in your global land cover analysis. This same logic would hold for plots that are fully forested but may be adjacent to a forest edge that is not captured on plot. It may be worth noting this possibility in the manuscript.

We appreciate the reviewer’s deep knowledge of the FIA methods. We have added a section to our methods document discussing these cases, explaining why we choose to exclude them, and the potential ramifications thereof (L244-252).

How were subplots where the mapped conditions changed between measurements treated?

In our expanded methods section we now describe how we treat changes in mapped conditions (L237-244).

While only trees ≥ 12.7 cm in diameter were included in the analysis, how were ingrowth trees, those they passed the diameter threshold between measurements treated in terms of BA and BAI estimates?

In-growth trees were included in both measurements. We discuss the implications of ingrowth in the methods section (L262-267).

While I appreciate that small diameter trees were not considered in the analysis based on the way edge was determined on subplots (micro plots may not have encompassed both forest and non-forest

conditions) it may be worth noting the role of small diameter trees in both the BA and BAI estimates in the paper. Again, if you conducted at the plot level you could consider the small diameter trees but if you restricted the analysis to the central subplot you could not.

We have added text to discuss the potential impact of excluding small diameter trees on our results (L81-83).

Basal area can be missing leading in terms of characterizing forest conditions, particularly forest structure. Including diameter distributions as a figure across the three edge-non-edge categories would help the reader understand if stand structure was markedly different between the categories. This would also lend support for the BAI estimates across categories - are edge forests smaller diameter (suggesting younger) resulting in higher BAI than "interior" forest which may be the same BA with fewer large diameter (older) trees but with very different BAI?

As described above, we are appreciative of this suggestion and have added a new figure (Fig. 2) showing diameter distributions and stem density distributions.

Given the study period and the remeasurement lengths for eastern FIA plots how did less than half of the forest-non-forest subplot conditions not have remeasurements? Was it because the map conditions changed between measurements? This needs further clarification to understand the differences. A table of n subplots by category would be helpful to understand samples sizes used in the regression analysis and would help, in part, to explain the CI in Figure 1 - sample size is likely a factor (following up on this as later in the methods this is acknowledged).

As mentioned in previous comment, our re-analysis has corrected the issue with re-measurements.

We have also included Table S1 with the sample sizes by category.

Matching and linear regressions

This is an interesting analysis and use of the NFI data. That said, some of the points I raised above may influence the matching of edge and interior plots.

We have re-done our matching as part of the revisions with alterations based on reviewer comments (including selection of only a single subplot per plot). We describe our revised matching procedure in the methods section (L284-287).

Mortality and timber harvest

As before, how might including small diameter trees < 12.7 cm dbh contribute to these results. Also, the FIA program provides estimates of net annual growth (which includes mortality and damage) and net change (which includes mortality, damage, and harvest removals) based on individual trees which might be considered as an alternative to BAI for estimates of biomass growth or carbon sequestration.

We have expanded our analysis of mortality and harvest removals so that our results are more representative of biomass growth. We have added discussion in the methods on the possible impacts of excluding small diameter trees on our mortality results (L361-363).

Reviewer #3 (Remarks to the Author):

This manuscript characterises the impacts of edge effects on forest structure and growth in North Eastern USA, using data on a national forest inventory. The authors use a linear model to analyse the data and make predictions about the impacts of forest edges across the region. The authors conclude that forest edges enhance both basal area and basal area and that this varies across different forest types. In addition, anthropogenic edges resulted in larger enhancements of basal area and basal area increment. Using these models to scale up to a regional level, the authors concluded that edges resulted in ~ 6% increase in BAI across forest in the North Eastern USA. This paper presents a novel result and is generally well written. However, I do have some doubts about the robustness of the methods used by the authors that I would like them to address.

Please note that my comments focus mainly on the problems that I perceive with the paper. This is simply due to my time constraints and not a reflection on the work. I try to focus on what I think will help to improve the paper most, but spend less time mentioning all the things that are great about the paper.

We thank the reviewer for their kind words!

Major issues

In the section on statistical matching, the authors state that ‘Matching successfully eliminated differences in all covariate distributions in both datasets.’ However, in their review of matching methods Schleicher et al (2020) state that ‘...in most instances matching reduces—but does not eliminate—differences between treatment and control units.’ As such, I would like to see evidence of how much difference there was between covariates used for matching both prior to and after statistical matching was undertaken. Schleicher et al (2017) provide a potential way in which to present this, which you can find in their supplementary materials.

We thank the reviewer for their comment and their suggestion of how to address it. We have added two tables (Tables S4 and S5) sensu Schleicher et al (2017) to present the efficacy of matching at reducing differences in covariate distributions.

In the materials and methods section, the authors describe using a linear regression model to estimate differences in basal area (BA) or basal area increment (BAI) between edge and core forest areas. Did the authors do any model simplification as part of this? The aim of any linear modelling should be to produce the simplest model that provides a good fit for the data. Without this, it isn't clear that edge and forest type are important predictors of BA or BAI in the current study. The authors should redo this analysis and include some form of model selection/simplification.

We have re-done our model analyses and expanded the related section in the methods text. We now include model selection using AIC and Residual Deviance as metrics for parsimony and goodness-of-fit. We report the results of this process in Table S2.

Regarding the modelling of BA and BAI, it would also be a good idea to examine any potential spatial autocorrelation that may be driving apparent relationships. At the very least, the authors should produce a figure to show how the residuals of their model vary in space.

We appreciate the reviewer's suggestion and included in our re-analysis a test for spatial autocorrelation. We have addressed this, along with the comments from reviewer #2, by excluding subplots from within the same FIA plot. We describe this process in the methods section (L280 – 287).

Minor issues

Line 46 - Best not to use the word ‘quinquennial’ here as nobody will know what it means, better to say ‘...collects measurements every five years...’ instead.

We have altered our language in the interest of accessibility and clarity.

Line 48 - BA is not exactly an analog of biomass since biomass calculations usually incorporate height as well. I usually just describe BA as a measure of forest structure.

We have added text to clarify that BA is a correlate of biomass, and have emphasized its relevance to forest structure.

Figure 1 - I'm not clear why, for the model that examines the effects of anthropogenic effects, you don't also have a result for edges that were 'natural.' From the methods section, it seems that you classified edges into these two groups so I would expect to see the results for these groups reported here.

We corrected the text in the methods document and clarified the limitations of our edge classification.

Lines 84-89 - Say which regions this relates to. As someone who isn't from North America this wasn't clear to me.

We added a figure in the SI (Figure S6) to better locate our results and make the paper more accessible to an international audience.

Lines 129-130 - The authors say that they quantified fragmentation in both tropical and temperate forest biomes, referring to Figure S6. However, Figure S6 is a comparison of fragmentation between different remote sensing products for the North Eastern USA. The authors should correct this.

We corrected the figure citation and added a new analysis to quantify the raw difference in edge-area between tropical and temperate forests (L405 – 409).

Figure 3 - It is quite hard to make out where temperate regions are on this map as the line is quite thin. Maybe this data would benefit from being a composite figure with a plot next to it showing the mean/median fragmentation of each ecoregion as well as the range of values in temperate vs tropical regions.

We appreciate the reviewer's suggestion and added a panel to Figure 4 that shows the area of fragmentation within each continent for both temperate and tropical forests.

In the methods section (Scaling edge effects on forest growth across the Northeast) the authors detail aggregation of the potential effects of edges on forest growth at the ecoregion level. What was the rationale for doing this at the ecoregion scale instead of aggregating using a finer scale grid? I worry that this aggregation may hide some of the inevitable within ecoregion variation. Similarly, I'm not clear why temperature, light, water, and nitrogen deposition were averaged across ecoregions as it is clear from figure S3 that you have relatively high resolution data on this.

We thank the reviewer for their comment and have added text to justify the use of ecoregions as the spatial unit (L128-130). We aggregated to the ecoregion level in an effort to reduce the impact of mismatches in spatial resolution between our gridded inputs. The light/water/temperature limitation datasets are gridded quite coarsely and would require aggregation to a similar spatial scale to the ecoregion. Ecoregions are a less arbitrary unit than the spatial grid cells and the ecoregion level that we selected are defined with similar biophysical properties to our predictor variables, limiting the within-ecoregion variation.

Reviewer comments, second round review –

Reviewer #1 (Remarks to the Author):

The authors have put a lot of effort into answering the reviewers' comments. The text has been revised and now offers more detail on the results. The authors have taken all my comments into account. Especially the focus is now more clearly shown (temperate forest, e.g. in the title). Some figures have been revised (Fig.2 and Fig.4) - Figure 2 shows new results and thus gives a deeper understanding (like more trees in the edge area) and Figure 4 now clearly shows the differences between tropical and temperate forests. My main criticism was the unclear statistics. This part has been extensively revised and expanded. From my point of view everything is explained sufficiently and comprehensibly.

Reviewer #2 (Remarks to the Author):

The authors have addressed my comments and concerns and the manuscript is markedly improved. While details of the definitions used for edges are included, to some extent, in the methods, including abbreviated versions of these early in the introduction would be helpful. See below for specific comments.

L3: Consider changing “understandings” to “understanding”

L20: What is an “anthropogenic forest edge”?

L25: Defining forest edges as distinct ecosystems here rather than a component of forest along a continuum of non-forest, forest edge, interior forest. Perhaps it is better to refer to these forest edges as fragments as you did later in this sentence if this is what you mean rather than the margin of a larger population which includes interior forest as you define it.

L42: As before regarding how you characterize forest edges. It would be helpful if you define how you classify forest edge/fragments early in the manuscript so that readers understand what you are referring to relative to intact forest, interior forest, non-forest...you might consider moving the short description on L66 up to the beginning of the introduction and to the abstract. The challenge if this is the criteria used to characterize forest edge is that these may indeed be the margins of intact forest so they do not necessarily represent forest fragments and given that the FIA data are spatially explicit but not spatially continuous it is not possible (without harmonizing with auxiliary forest cover data) to assess if the forest edges are also forest fragments. Clarifications of these points early in the text may be sufficient.

L77: Again, it would be useful to define: “interior forest”

Reviewer #3 (Remarks to the Author):

Thank you for your helpful responses to the issues I previously raised in my review. However, I still have a few issues that I think need to be corrected.

1. For the presentation of the model results (Table S2) I recommend using a pseudo R² value rather than residual deviance as it is easier to interpret. I think that you should also give pseudo r² values in the main text to make it explicit how well (or poorly) the models fit. I think being honest about model fit is really important so that we don't run the risk of overselling results.
2. Figure 4 needs to be labelled as (a) and (b) - so that it matches the legend.
3. In the supplementary materials the headings for the tables should go above the tables, rather than below them.
4. For Table S2, the standard way to present models is with the 'best' model at the top and the less parsimonious models below this.
5. Regarding the data availability, the data should be placed on a dedicated repository using something like zenodo, dryad, or figshare. Links to dropbox folders etc often end up failing after a few years.